# Fucoidan-Rich Substances from *Ecklonia cava* Improve Trimethyltin-Induced Cognitive Dysfunction via Down-Regulation of Amyloid β Production/Tau Hyperphosphorylation

**DOI:** 10.3390/md17100591

**Published:** 2019-10-17

**Authors:** Seon Kyeong Park, Jin Yong Kang, Jong Min Kim, Seul Ki Yoo, Hye Ju Han, Dong Hwa Chung, Dae-Ok Kim, Gun-Hee Kim, Ho Jin Heo

**Affiliations:** 1Division of Applied Life Science, Institute of Agriculture and Life Science (BK21 plus), Gyeongsang National University, Jinju 52828, Korea; tjsrud2510@naver.com (S.K.P.); kangjy2132@naver.com (J.Y.K.); myrock201@naver.com (J.M.K.); ysyk9412@naver.com (S.K.Y.); gksgpwn2527@naver.com (H.J.H.); 2Food Technology Major, Graduate School of International Agricultural Technology, Institute of Green Bio Science and Technology, Pyeongchang 25354, Korea; dchung@snu.ac.kr; 3Department of Food Science and Biotechnology, Kyung Hee University, Yongin 17104, Korea; DOKIM05@khu.ac.kr; 4Departments of Food and Nutrition, Duksung Women’s University, Seoul 01369, Korea; ghkim@duksung.ac.kr

**Keywords:** *Ecklonia cava*, trimethyltin, cognitive function, amyloid beta, tau

## Abstract

*Ecklonia cava* (*E. cava*) was investigated to compare the effect of polyphenol and fucoidan extract and mixture (polyphenol:fucoidan = 4:6) on cognitive function. The ameliorating effect of *E. cava* was evaluated using the Y-maze, passive avoidance and Morris water maze tests with a trimethyltin (TMT)-induced cognitive dysfunction model, and the results showed that the fucoidan extract and mixture (4:6) had relatively higher learning and memory function effects than the polyphenol extract. After a behavioral test, the inhibitory effect of lipid peroxidation and cholinergic system activity were examined in mouse brain tissue, and the fucoidan extract and mixture (4:6) also showed greater improvements than the polyphenol extract. Mitochondrial activity was evaluated using mitochondrial reactive oxygen species (ROS) content, mitochondrial membrane potential (MMP, ΔΨm), adenosine triphosphate (ATP) content, and mitochondria-mediated protein (BAX, cytochrome C) analysis, and these results were similar to the results of the behavioral tests. Finally, to confirm the cognitive function-related mechanism of *E. cava*, the amyloid-β production and tau hyperphosphorylation-medicated proteins were analyzed. Based on these results, the improvement effect of *E. cava* was more influenced by fucoidan than polyphenol. Therefore, our study suggests that the fucoidan-rich substances in *E. cava* could be a potential material for improving cognitive function by down-regulating amyloid-β production and tau hyperphosphorylation.

## 1. Introduction

Neurodegenerative diseases, such as Alzheimer’s disease (AD) and Parkinson’s disease (PD), are predicted to be the second-most common cause of death following cancer among the elderly by the 2040s [1]. AD is a progressive neurodegenerative disease that is characterized by two classic hallmarks (amyloid plaque and neurofibrillary tangles (NFTs)), leading to cognitive deficits and learning and memory impairment [2]. Oxidative stress has long been regarded as a major factor in neurodegenerative diseases, and abnormally increased oxidative stress induces amyloid beta (Aβ) production and accumulation in brain tissue by activating β-secretase and the imbalance of production and clearance [3,4]. The formation of NFTs, which arises from the misfolding of hyperphosphorylated tau in synaptic nerve terminals, causes synaptic dysfunction and then causes neuronal cell death. For this reason, many synthetic neuroprotective agents have been used; however, synthetic materials have been reported to have certain side effects, such as dry mouth, tiredness, drowsiness, sleepiness, anxiety, and nervousness [1]. Hence, researchers have an increasing interest in safe and effective agents for neurodegenerative diseases. 

TMT, which is an organotin compound with a neurotoxic substance, induces behavioral changes with massive neuron loss and neuroinflammation in rodent hippocampus [5]. The neurotoxic action of TMT leads to a cytotoxic response by triggering a series of molecular events and cellular pathways via the activation of various kinases, such as c-Jun N-terminal kinase (JNK) and protein kinase C, and transcription factors, such as nuclear factor kappa B (NF-κB) and stress proteins [6]. In particular, TMT causes cognitive dysfunction by inducing selective damages in the hippocampal CA1 and CA3 regions [7]. Therefore, TMT is considered to be a suitable substance for the study of cognitive impairment in an in vivo model.

The oceans occupy more than 70% of the world’s surface, and 90% of living biomass, comprising approximately half of the total world’s species, is found in oceans [1,8]. Edible seaweeds are reported to be an abundant bioactive source of carbohydrates, soluble dietary fibers (pectic substances, algal polysaccharides, part of the hemicelluloses), essential amino acids, vitamins (A, B1, B2, B3, B6 and C) and minerals (iodine, potassium, iron, magnesium, and calcium) with raw caloric value [8,9,10]. Although marine algae are a potential source of new bioactive substances with low toxicity for various diseases, research on them has been relatively insufficient. In particular, *E. cava*, which is one of the brown algae, is evaluated as an important source due to its valuable health beneficial effects by including a variety of biological compounds such as peptides, polysaccharides, carotenoids, fucoidan and phlorotannins [10]. In particular, polyphenol and fucoidan have been reported as the major bioactive substances of *E. cava*. Phlorotannins, which are a major polyphenol found only in marine brown algae, are composed of the polymerization of phloroglucinol units (e.g., eckol, dieckol, 6,6′- bieckol, and eckstolonol) [10]. Phlorotannin derived from *E. cava* has been extensively studied for its anti-inflammatory, anti-allergy, anti-diabetic, and anti-cancer effects based on its strong antioxidant activity [10,11]. Fucoidan, which is a class of sulfated fucose-rich polysaccharides from brown algae, has been reported to have anticoagulant and anti-thrombotic, anti-virus, anti-tumor, anti-inflammatory, blood lipid reduction, and gastric protective effects [12,13]. Therefore, we tried to evaluate the cognitive improvement effect of *E. cava* according to a comparison between polyphenol extract, fucoidan extract, and their mixture (polyphenol: fucoidan = 4:6), which was selected based on their antioxidant effect and neuronal cell protective effect (Appendix A). Briefly, the antioxidant effect was evaluated by measuring the ABTS/DPPH radical scavenging activity and inhibitory effect of lipid peroxidation, and the results showed strong antioxidant effects by increasing the ratio of polyphenol (Appendix A). An inhibitory effect against acetylcholinesterase (AChE) was also showed to occur by increasing the ratio of polyphenol (Appendix A). In addition, the cell protective effect was evaluated using intracellular reactive oxygen species (ROS) content and cell viability on H_2_O_2_-induced neuronal cells (PC-12 and MC-IXC cells), and the results exhibited cytotoxicity when the ratio of polyphenol to fucoidan was five or more (Appendix A). As a result, a mixture of polyphenol and fucoidan could be a more effective treatment for protecting neuronal cells than other extracts (including polyphenol or fucoidan), and the final ratio was selected as 4:6 (polyphenol:fucoidan). Based on these results, we intend to evaluate and develop the possibility of a substance for industrial use of the mixture (polyphenol:fucoidan = 4:6). Therefore, the cognitive-enhancing effect of the mixture from *E. cava* was evaluated and compared with two extracts (including polyphenol and fucoidan) on a TMT-induced cognitive dysfunction mouse model.

## 2. Results and Discussion

### 2.1. Behavioral Tests

To confirm the ameliorating effect of the *E. cava* (polyphenol/fucoidan extract and mixture (4:6)) on TMT-induced learning and memory impairment, Y-maze, passive avoidance, and Morris water, maze tests were conducted. TMT causes learning and memory impairment by inducing selective damages in the hippocampal CA1 and CA3 regions [14]. Hippocampal damage leads to learning and memory impairment and behavioral changes [6].

The spatial cognitive function was evaluated using the Y-maze test, and the results are shown in Figure 1A,B. The spatial cognitive function of mice was impaired by a TMT injection, and the results showed that spontaneous alternation behavior of the TMT group (30.97%) decreased approximately 9.52% compared to that of the control group (40.49%) (Figure 1A). The administration of the fucoidan extract (38.61%) and mixture (4:6; 33.03%) showed slightly improved spontaneous alternation behavior in contrast to the polyphenol extract (27.73%). In contrast, Y-maze results showed a similar number of total arm entries and indicated no differences in overall behavioral activity among all groups (Figure 1A). In Figure 1B, the 3D image shows the path tracing of mice during the Y-maze test. While the control group exhibited similar movement in all arms, the TMT group showed increasing movement in a specific arm as a result of damage to the spatial cognitive function. The fucoidan extract and mixture (4:6) groups showed movements similar to those of the control group.

To evaluate the short-term learning and memory function, a passive avoidance test was conducted (Figure 1C). The results indicated significant learning and memory impairment by TMT injection; the TMT group (20.60 s) showed a 90.52% decreased step-through latency time compared to that of the control group (217.40 s). The fucoidan extract (156.60 s) and mixture (4:6; 135.40 s) effectively improved short-term learning and memory function compared to the polyphenol extract group (36.20 s).

Long-term learning and memory ability were evaluated using a Morris water maze (Figure 1D–F). In the training (hidden platform) session, all groups exhibited a decreased escape latency time for four days (Figure 1D). In the 4-day training session, the TMT group (38.38 s) showed a relatively high escape latency time compared to that of the control group (26.06 s). The administration of polyphenol extract (35.50 s), fucoidan extract (30.87 s), and mixture (4:6; 35.05 s) showed a similar escape latency time. However, in the probe session, the polyphenol extract (26.71%), as well as the fucoidan extract (31.73%) and mixture (4:6) group (29.87%) showed significant learning and memory function improvement for the platform learned as an escape space compared to the TMT group (22.89%), and these results were statistically significant with the control group (28.17%) (Figure 1E). In Figure 1E and F, the images show the path tracing of mice and learning and memory ability about the platform in the W zone. The control group showed relatively more movement in the W zone, where the platform existed. However, the TMT group showed decreased movement in the W zone, and the fucoidan extract group and mixture (4:6) groups showed a tendency to increase movement in the W zone.

In our three behavioral tests (Y-maze, passive avoidance, and Morris water maze), the cognitive improvement effect of *E. cava* is considered to be more influenced by fucoidan than polyphenol. According to Lee et al. [15], fucoidan effectively improved behavior ability in the passive avoidance test and Morris water maze tests in scopolamine-induced learning and memory impairment Sprague-Dawley rats. These memory improvement effects were reportedly due to the suppression of inflammatory responses by decreasing COX-2 mRNA expression and TNF-α and IL-1β. In addition, fucoidan effectively restored the expression of memory-related mRNA expression, such as brain-derived neurotrophic factor (BDNF) and cAMP-response element-binding protein (CREB) in the hippocampus. Based on these reports, the fucoidan extract from *E. cava* effectively ameliorated behavioral changes by restoring TMT-induced hippocampal degeneration in brain tissue.

### 2.2. Inhibition of Lipid Peroxidation in Mouse Brain Tissue

Brain tissue is very vulnerable to oxidative stress because it has a very high content of unsaturated fatty acids relative to other organs [16]. In particular, polyunsaturated fatty acids, including carbon-carbon that double bonds, easily are attacked to oxidants such as free radicals and ROS. As a result of lipid peroxidation, malondialdehyde (MDA) is produced in various organs, such as liver, heart, and brain tissue, and its adducts can participate in deleterious secondary reactions [17]. Therefore, MDA content is a useful marker of endogenous lipid peroxidation in neurodegenerative diseases [9,18].

The inhibitory effect of lipid peroxidation stimulated by the TMT injection was evaluated by measuring the MDA content in mouse brain tissue (Figure 2). As a result, the TMT group showed increased MDA content (1.27 nmole/mg of protein; 119.81% increase) compared to the control group (1.06 nmole/mg of protein). In contrast to the TMT group, fucoidan extract (1.08 nmole/mg of protein) and mixture (4:6; 1.12 nmole/mg of protein) effectively inhibited MDA production in mouse brain tissue.

Although phlorotannins as major polyphenols from marine algae have been reported to have excellent antioxidant potential, our results showed a more effective inhibitory effect of fucoidan extract on lipid peroxidation than polyphenol extract. There is still not enough research on the antioxidant effects of fucoidan, and it has not yet been demonstrated. In particular, research on the antioxidant effects of *E. cava* has been focused on polyphenol, and research on fucoidan is rarely found. Furthermore, the antioxidant effects of fucoidan extract from *E. cava* were not detected (Appendix A) in an in vitro assay. Fucoidan, which is composed of fucose-containing sulfated polysaccharides, is a cell wall component of brown seaweed [19]. The antioxidant activity of sulfated polysaccharides is known to be determined by their structural features, such as degree of sulfating, molecular weight, type of the major sugar, and glycosidic branching [1]. In particular, the high sulfate content of their structure exhibits a strong radical scavenging effect, reducing power, and chelating power [20]. Moreover, the sulfated polysaccharide fraction F2 (850 kDa) from *Porphyra haitanesis* containing 20.5% sulfate content showed an antioxidant effect in vivo by increasing the antioxidant enzyme activity (superoxide dismutase (SOD) and GPH-Px) and total antioxidant capacity in various organs (lung, liver, heart, spleen, serum, and brain tissue) of aging Kunming mice (20 months old) [18]. It effectively decreased MDA content, which significantly increases with aging in all the organs. Based on these studies, fucoidan from *E. cava* is also considered to be a potential material for affecting biological activities with potential health benefits. Thus, the inhibitory effect on lipid peroxidation in the fucoidan and mixture (4:6) groups leads to a conjecture that it acts through an indirect pathway by its metabolites rather than the direct effect of fucoidan.

### 2.3. Cholinergic System Activity 

The cholinergic system of the brain is closely related to cognitive function. Cholinergic signaling is initiated by the release of acetylcholine (ACh) as a neurotransmitter across a synapse between neurons and released ACh carries signals by binding with ACh receptors (e.g., muscarinic or nicotinic receptors) [21]. However, in AD patients, there is a decrease in ACh content due to mitochondrial dysfunction from oxidative stress. ACh is formed by choline combined with acetyl-CoA synthesized from mitochondrial pyruvate in cholinergic neurons [22]. Cognitive decline then occurs as neurotransmission by ACh decreases. Therefore, AChE inhibitors are suggested as the major therapeutic agents for AD by inhibiting the hydrolysis of ACh in the synapses of neurons, and cholinergic therapies based on AChE inhibition have resulted in well-proven improvement in AD patients [23,24]. 

The results of AChE activity, AChE expression level, and ACh content as the major markers of the cholinergic system are shown in Figure 3. In Figure 3A, AChE activity of the TMT group as the result of an impaired cholinergic system was increased by approximately 113.82% compared to the control group (100.00%). The fucoidan extract group (101.76%) had significantly inhibited AChE activity, similar to that of the control group, and the mixture (4:6) group had slightly reduced AChE activity in contrast to the polyphenol extract group (113.76%). The AChE expression level was increased by the TMT injection, and the fucoidan extract effectively reduced AChE expression compared to the TMT group, and the mixture group also slightly reduced AChE expression (Figure 3B). Finally, in Figure 3C, increased AChE activity and expression levels were induced to decrease the ACh content in brain tissue in the TMT group (3.33 mmole/mg of protein; 20.33% decrease) compared to that of the control group (4.18 mmole/mg of protein). On the other hand, fucoidan extract effectively increased the ACh content in mouse brain tissue compared to the TMT group by inhibiting AChE activity and the expression level. The mixture group (3.51 mmole/mg of protein) showed slightly improved ACh content compared to the polyphenol extract group (3.27 mmole/mg of protein).

According to a previous study, TMT treatment remarkably attenuated the cholinergic system by increasing AChE activity in the dentate gyrus of the hippocampus as well as reducing choline acetyltransferase (ChAT) activity, which is a transferase enzyme responsible for the synthesis of ACh [23]. Phlorotannins, major polyphenol compounds in the brown seaweed family, include dieckol, 6,6′-bieckol, eckol, and eckstolonol [10]. And another study reported that the phlorotannins from the *Ecklonia* species acted as a strong AChE and butyrylcholinesterase inhibitor in an in vitro assay [25,26]. Phlorotannins from *Ecklonia stolonifera,* such as eckstolonol (42.66 mM), eckol (20.56 mM), phlorofucofuroeckol-A (4.89 mM), dieckol (17.11 mM), 2-phloroeckol (38.13 mM), and 7-phloroeckol (21.11 mM) [26] effectively inhibited AChE with 50% inhibition concentration (IC_50_) values. In our study, the polyphenol extract showed a significant inhibitory effect against AChE derived from PC-12 cells and had an IC_50_ value of 70.63 μg/mL (Appendix A). 

In contrast to these in vitro AChE tests, polyphenol extract, including dieckol as a representative material, did not effectively inhibit AChE activity in a TMT-induced cholinergic dysfunction mouse model. The absorption and bioavailability of the fucoidan/polyphenol extract or their metabolites are considered to be higher than fucoidan in brain tissue of the in vivo model, and further studies on their absorption mechanism are needed. Therefore, our results suggest that fucoidan-rich substances in *E. cava* could be a more suitable material than polyphenol for the treatment and prevention of learning and memory dysfunction by enhancing the cholinergic system through the AChE inhibitor and increasing ACh content. Giacobini et al. [27] also reported that cholinergic inhibitors could be helpful with amyloid plaque formation and neuronal cell death.

### 2.4. Mitochondrial Activity

TMT as an organotin compound has been used as a neurotoxin for cognitive dysfunction and neuron death because it selectively triggers apoptosis in specific subregions of the mammalian CNS [6,28]. TMT-induced apoptosis is directly associated with mitochondria by binding to stannin, which may mediate the selective toxicity of organotins as a membrane-bound protein found mainly in mitochondria [28,29]. In mitochondria, TMT combined with protein stannin could become demethylated with dimethyltin to cause damage to the integrity of the mitochondria membrane, and induce an apoptosis cascade with the release of cytochrome C and activation of caspases [5]. Therefore, the ameliorating effect on TMT-induced mitochondrial damage was evaluated by measuring the mitochondrial ROS content, mitochondrial ATP content, and mitochondria-mediated protein expression level (Figure 4). The mitochondrial ROS content was experimented on using DCF-DA dye, and the results are shown in Figure 4A. The TMT group showed significantly increased fluorescence intensity (187.98%; an increase of about 1.88 times) compared to the control group (100.00%). The administration of fucoidan extract (118.16%) and mixture (4:6; 136.25%) remarkably attenuated mitochondrial ROS content in contrast to the administration of polyphenol extract (165.00%). In Figure 4B, TMT injection-induced mitochondria membrane damage, and as a result, the TMT group showed decreased MMP (80.60%; about a 19.40% decrease) compared to the control group (100.00%). On the other hand, the fucoidan extract group showed significantly restored MMP (90.75%; about a 10.15% increase) compared to the TMT group. Finally, TMT-induced mitochondria dysfunction leads to a decrease in ATP production, and the TMT group (1.50 nmole/mg of protein; 45.05% decrease) decreased in mitochondrial ATP content compared to the control group (2.73 nmole/mg of protein). In particular, ATP content increased in the fucoidan extract groups (2.71 nmole/mg of protein) compared to the TMT group by restoring mitochondria function. The polyphenol extract and mixture (4:6) groups had slightly lower ATP content (2.00 and 2.25 nmole/mg of protein, respectively) compared to the fucoidan extract group (Figure 4C). Also, BAX and mitochondrial cytochrome C as mitochondria-mediated protein were measured, and the results showed that the fucoidan extract group inhibited BAX expression and prevented the release of cytochrome C from mitochondria to cytosol compared to the TMT group (Figure 4D–F).

Mitochondrial function deficits have been considered a major cause of neurodegenerative diseases such as AD, PD, and Huntington’s disease, and mitochondria function-enhancing substances have been proposed as therapeutic and prevention agents for AD [30]. Oxidative stress causes cell apoptosis through reduced energy metabolism by their structural collapse and oxidative damage. ROS is produced as a result of oxidative phosphorylation during mitochondrial energy metabolism at the ubiquinone site complex III of the mitochondrial electron transport chain [31]. However, the abnormally high levels of ROS lead to a decrease in MMP (ΔΨm), which ultimately reduces energy production by increasing mitochondrial membrane depolarization. In addition, the mitochondria are closely related to apoptosis due to pro/anti-apoptotic Bcl-2 family proteins and caspase activators [32,33]. Bcl-2 and Bcl-xL are anti-apoptotic proteins that function to prevent apoptosis by hydrogen peroxide or thiol depletion and to inhibit lipid peroxidation [31]. However, BAX, which is a member of the Bcl-2 protein family, is a pro-apoptotic protein regulator as resides, and the formation of BAX oligomeric pores in the outer mitochondrial membrane passes cytochrome C from mitochondria to cytosol [32]. The release of cytochrome C to cytosol triggers the activation of caspases by forming the apoptosome with apoptotic protease-activating factor 1 and then activating caspase –9, –3, and –7 induced neuronal apoptosis [33,34]. Therefore, the discovery of materials for mitochondria activation can maintain mitochondria homeostasis, which is critical to cognitive improvement by inhibiting neuronal cell apoptosis. Based on these reports, it might be possible to use fucoidan from *E. cava* as a therapeutic and preventive agent for cognitive-enhancing agents by blocking mitochondria dysfunction and neuronal cell apoptosis with the elevation of ATP levels, improving MMP (ΔΨm), and decreasing ROS content and downregulating BAX and the release of cytochrome C in treating TMT-induced mitochondria damage.

### 2.5. Cognitive Function-Related Mechanism

The major causes of cognitive dysfunction are known to be the accumulation and fibrillary deposition of the amyloid-β peptide in senile plaques and tau phosphorylation in NFTs [2]. JNK, a stress-activated protein kinase, is most implicated in AD [35]. JNK is activated and phosphorylated by reactive oxygen species (phosphorylated JNK; *p*-JNK), and *p*-JNK was detected in the neurons and dystrophic neurites of an AD model. The *p*-JNK elicits insulin resistance (IR) during binding to the insulin receptor substrate (IRS)-1 by inducing the phosphorylation of serine residues [36]. The phosphorylation of the IRS on serine residues leads to the inhibition of the insulin-degrading enzyme (IDE). IDE is known as an insulin-degrading enzyme, but it also acts to degrade the Aβ peptide. Finally, decreased IDE inhibits Aβ clearance and leads to neuronal loss and cognitive dysfunction with the accumulation and aggregation of Aβ peptides in brain tissue [3,4]. In addition, Aβ accumulation causes a cascade action of JNK/IRS-1, and the activation of the JNK signaling pathway is related to tau hyperphosphorylation via the PI3K/Akt/GSK-3 signaling pathway. PI3K/Akt is known as a neuronal growth and survival-related protein, and phosphorylation of PI3K/Akt can also activate GSK-3 (*p*-GSK-3) as an upstream regulator of GSK-3, which directly regulates tau phosphorylation. GSK-3 isomers (α/β) play an important role in the formation of axons by phosphorylating serine residues (*p*-GSK-3α(Ser21)/β(Ser9)) [37]. Therefore, inactivated GSK-3 induces abnormal tau phosphorylation by reducing the microtubule combination. As a result, hyperphosphorylated tau leads to learning and memory impairment by inducing neuronal cell damage and apoptosis with the development of NFTs.

Accordingly, the cognitive improvement mechanism of *E. cava* was evaluated by measuring the amyloid-β production/Tau hyperphosphorylation mechanism-related protein expression level (Figure 5). The TMT group exhibited increased *p*-JNK and *p*-IRS (ser307) (Figure 5A–C) and as a result, induced the inhibition of IDE activity (Figure 5A,D). The decreased expression of IDE increased the expression level of Aβ by inhibiting Aβ clearance (Figure 1E) in the TMT group compared to the control group. The administration of fucoidan extract and mixture (4:6) effectively reduced the expression level of Aβ by regulating the JNK/IRS-1/IDE signaling pathway. In addition, the TMT group showed a decreased expression level of *p*-Akt and *p*-GSK-3 by Aβ accumulation, resulting in tau hyperphosphorylation (Figure 5A,F–H). On the other hand, the fucoidan extract and mixture (4:6) groups showed a significant decrease in the expression level of *p*-tau by up-regulation of Akt/GSK-3. Based on these results, the cognitive function improvement effect of *E. cava* was influenced more by fucoidan than polyphenol as a bioactive substance.

In the amyloid cascade hypothesis, a typical mechanism related to cognitive dysfunction, the Aβ peptide is one of the major hallmarks of AD. Aβ production induces the activated JNK/IRS-1/IDE signaling pathway by oxidative stress stimulation [36]. PI3K/Akt is also associated with amyloid precursor protein excretion and IDE activity, and as a result, causes Aβ cascade production [25]. Aβ accumulation and aggregation lead to synaptic dysfunction, tau hyperphosphorylation, and neurodegeneration [35]. The formation of amyloid plaques with the aggregation of the Aβ peptide is more neurotoxic, and the Aβ peptide is associated with various cellular mechanisms [25]. Wei et al. [38] reported that fucoidan inhibited cell apoptosis by regulating the caspases (–3, –8, and –9) and antioxidant-related factors (SOD and glutathione) in damaged PC-12 cells by Aβ_25–35_ and D-galactose, and also improved learning and memory impairment in the Aβ_25–35_ and D-galactose-induced AD model. Moreover, fucoidan derived from *Laminaria japonica* effectively ameliorated animal behavioral abilities in an Aβ-infused learning and memory impairment rat model due to the regulation of the cholinergic system (ACh, ChAT contents, and AChE content) and antioxidant activity (SOD, GSH-Px, and MDA content) in the hippocampus of mouse brain [24]. Also, it inhibited cell apoptosis by regulating mitochondria-related proteins (Bcl-2/BAX ratio and caspase-3 activity). These reports suggest that fucoidan may be an excellent neuroprotective agent against Aβ-induced learning and memory impairments.

NFTs, which are insoluble twisted fibers, were found in neuronal cells and consist of phosphorylated tau. Tau is part of a structure called a microtubule that transports nutrients and other important substances in the nerve cell. However, in AD patients, the hyperphosphorylated tau protein is released by microtubule structures that collapse. In the current hypothesis and theories on the pathogenesis of AD, IR is reported as a key factor. The expression of glucose transporters and the insulin signaling genes determine tau and Aβ pathology in brain tissue [35]. Also, when the expression level of insulin signaling-related proteins such as IRS-1 and Akt is low, IR promotes the formation of NFTs by inactivating GSK-3, which is known to be closely related to tau hyperphosphorylation. According to Ali et al. (2015) [39], the injection of Aβ_1-42_ induced tau hyperphosphorylation, by inactivating GSK-3β through the inhibition of *p*-PI3K/Akt as an upstream regulator in C57BL/6N mice, and these mechanisms were ameliorated by the administration of melatonin, which is known as a hormone that regulates the sleep-wake cycle and is attracting attention because of its strong antioxidant activity and neuroprotective effect. In addition, these results demonstrated that the inhibition of Aβ-mediated tau hyperphosphorylation increased the number of survival neurons in the dentate gyrus, CA3, and CA1 regions of the hippocampus.

The results demonstrated that the cognitive-enhancing mechanism of *E. cava* might be attributed to regulating Aβ production by regulating the JNK/IRS-1/IDE signaling pathway and tau hyperphosphorylation by up-regulation of the Akt/GSK-3 signaling pathway with the action of fucoidan-rich substances (fucoidan extract and the mixture (polyphenol: fucoidan = 4:6)) on brain tissue. That is, the regulation of major pathophysiologic pathways (cholinergic system, mitochondrial activity, and Aβ/tau-related mechanism) of AD suggests the possibility of treatment and prevention for cognitive decline. Based on these results, the possibility of using the mixture (polyphenol:fucoidan = 4:6) from *E. cava* in medicine as a cognitive-enhancing substance was confirmed in the TMT-induced cognitive dysfunction mouse model, and the enhancement effect was more influenced by fucoidan than polyphenol.

### 2.6. Chemical Composition and Molecular Weight of Fucoidan from E. cava

The average molecular weight and chemical composition (sulfate and monosaccharide composition) of fucoidan extract from *E. cava* are shown in Table 1. The sulfate content of fucoidan was 3.03%, and monosaccharide of fucoidan was shown to be composed of arabinose (2.03%), fucose (15.19%), galactose (9.62%), glucose (11.42%), rhamnose (1.44%), xylose (23.77%), and other monosaccharides (36.53%). The average molecular weight of fucoidan was analyzed to be 110.78 kDa by GPC-LC after calibration with a molecular weight standard marker (Table 1).

Fucoidan is commonly found in brown seaweed and is composed of sulfated heteropolysaccharides [12]. Also, different monosaccharide contents, glycosidic linkage, sulfate content, and their structure have been reported to characterize bioactive reactions [40]. Most studies on fucoidan have reported that their bioactivity is mainly anticoagulant and anti-thrombin activity due to its unique chemical structure, which has a heparinoid compound [41]. On the other hand, recent studies have reported other bioactivities, such as anti-virus, anti-tumor, anti-complementary, and immunomodulatory activity [42]. In particular, some neuroprotective effects and immunological activity of fucoidan have been reported. The fucoidan from *Laminaria japonica* showed a neuroprotective effect on H_2_O_2_-induced apoptosis in PC-12 cells by activating the PI3K/Akt pathway, which is closely related to cell survival [43]. Aβ, as a major factor of AD, has been implicated as neuroinflammatory, which then induced cognitive dysfunction [2]. Toll-like receptors (TLR), which is a class of membrane-spanning receptors, is related to neurodegeneration in CNS, including astrocytes, neurons, and oligodendrocytes, by leading to a neuroinflammatory reaction [44]. In particular, Liu et al. [45] demonstrated that TLR-2 is a primary receptor for Aβ, and it related to an Aβ-induced inflammatory reaction and Aβ phagocytosis in cultured microglia and macrophages. Makarenkova et al. [46] reported that fucoidan from brown seaweeds (*Laminaria japonica*, *Laminaria cichorioides,* and *Fucus evanescens*), which have a different structure, could improve immune response by interacting with TLR (–2 and –4) as TLR ligands in human embryonic kidney cells (HEK293-null, HEK293-TLR2/CD14, HEK293-hTLR4/CD14-MD2, and HEK293-hTLR5). The absorption mechanism of fucoidan through the gastrointestinal tract is unclear. However, some reported studies suggest that polysaccharides of brown seaweed could act as prebiotics and inflammatory-related signaling molecules due to easily fermentable materials by the intestinal microbiota such as *Lactobacillus* strains [47,48]. The fermentation of prebiotics with intestinal bacteria stimulates an immune response by stimulating the activity and growth of beneficial bacteria and the production of short-chain fatty acids [47]. In addition, prebiotics regulate a neuropeptide (peptide YY) as gut hormones, hippocampal BDNF, and the subunit of N-methyl-D-aspartate receptor [48]. As the results show, prebiotics could affect brain function, such as anxiety, depression, stress, autism, learning, and memory through the modulation of gut microbiota–brain axis by improving the intestinal microbiota [48]. Along with these reports, our results also showed that the fucoidan-rich substances in *E. cava* downregulated Aβ production and tau hyperphosphorylation. Therefore, fucoidan-rich substances in *E. cava* might be useful as therapeutic and preventive agents for neurodegenerative diseases.

## 3. Materials and Methods

### 3.1. Chemicals

Acetylthiocholine, 5,5-dithiobis (2-nitrobenzoic acid), TMT, thiobarbituric acid, trichloroacetic acid, phosphoric acid, digitonin, triton X-100, and all other chemicals used were purchased from Sigma-Aldrich Chemical Co. (St. Louis, MO, USA). 

### 3.2. Sample Preparation

*E. cava* was collected from July to September 2017 in Yewol, Jeju Island, Korea. *E. cava* was washed with purified water three times repeatedly to remove salt and air-dried for 12 h at 50 ± 5.0 °C.

#### 3.2.1. Preparation of Polyphenol Extract from *E. cava*

Dried *E. cava* powder was crushed and extracted with ethanol under reflux conditions. The extract was filtered with a centrifuge, and the supernatant was collected. Filtrates were evaporated under vacuum, dried, and used as an *E. cava* polyphenol extract. The yield of *E. cava* polyphenol extract was 8.0% (w/w). According to a previous report, *E. cava* is reported to be a richer source of phenolic compound contents than other seaweeds, and the isolated and characterized major substances are phlorotannins, including 7-phloroeckol, dieckol, and 6,6-bieckol [11]. Dieckol content, as a reference compound of *E. cava* polyphenol extract, was analyzed using the HPLC system (Waters, MA, USA) [49]. Dieckol content of *E. cava* polyphenol extract was detected to be 4.8% (w/w) (Table 2)

#### 3.2.2. Preparation of Fucoidan Extract from *E. cava*


Dried *E. cava* powder was extracted with HCl in acid hydrolysis. The hydrolysate was filtered using a decanter and then the supernatant was collected. The supernatant was evaporated under vacuum, and ethanol was added to the extract and soaked. The extract was filtered with a continuous centrifuge, and the pellet was collected. The pellet was lyophilized, and then an *E. cava* fucoidan extract was obtained. The yield of *E. cava* fucoidan extract was 1.8% (w/w).

#### 3.2.3. Preparation of Mixture (Polyphenol:Fucoidan) from *E. cava*


The mixture was prepared by dissolving polyphenol extract and fucoidan extract in drinking water, respectively, and mixing them at a ratio of 4:6 (polyphenol:fucoidan).

### 3.3. In Vivo Experimental Design

#### 3.3.1. Animals 

Institute of Cancer Research (ICR) mice (male, 4 weeks) were obtained from Samtako (Osan, Republic of Korea). The mice were randomly assigned to three per cage and maintained with a 12 h light/dark cycle, 55% humidity, and 22 ± 2 °C. The mice were divided into six groups (*n* = 12), which consisted of a normal control (NC) group, TMT-injected group (negative control group), and sample groups (polyphenol extract, fucoidan extract, and mixture (polyphenol:fucoidan; 4:6) 20 mg/kg of body weight, respectively). Sample dosage for mice was determined on the basis of previous researches [50,51]. The groups were fed orally once a day for 3 weeks. After oral administration, the control group was injected with 0.85% (w/v) sodium chloride solution. All the other groups were injected with TMT solution (2.5 mg/kg; 7.6 μg/kg of body weight). All animal experiment methods were conducted according to the Institutional Animal Care and Use Committee of Gyeongsang National University (certificate: GNU-170605-M0023), and performed in accordance with the Policy of the Ethical Committee of the Ministry of Health and Welfare, Republic of Korea. 

#### 3.3.2. Y-Maze Test

The Y-maze consisted of three arms made of the same black-painted plastic (33 cm long, 15 cm high, and 10 cm wide). Each mouse movement was recorded by a smart 3.0 video tracking system (Panlab, Barcelona, Spain) for 8 min [51].

#### 3.3.3. Passive Avoidance Test 

The passive avoidance test box was divided into a light and dark zone. In the training trial, each mouse was allowed to move freely between the two zones for 1 min and received an electric shock (0.5 mA, 3 s) as soon as it entered the dark zone. After 24 h, in the test trial, the step-through latency time to the dark zone was measured (maximum time limit: 300 s) [51].

#### 3.3.4. Morris Water Maze Test 

The Morris water maze test was conducted, according to Morris (1984) [52], with some modifications. A stainless-steel circular pool (90 cm in diameter) was divided into quadrants (E, W, S and N zones) with visual clues on the wall for navigation and squid ink (Cebesa, Valencia, Spain) was added to the pool water (20 ± 2 °C). A platform (6 cm in diameter), which was a place to escape from the water, was placed in the center of the W zone, and its position was not changed during the training session. During the training sessions (days 1–4), the latency time for each mouse to reach the platform (maximum time: 60 s) was recorded, and they were pulled out of the pool when they stayed on the platform for 15 s. The trials were conducted four times a day for four consecutive days. In a probe test (day 5), the time spent in the W zone was recorded for 60 s after removing the platform.

### 3.4. Inhibition of Lipid Peroxidation

After the behavioral tests, the mice were sacrificed for biochemical studies, and whole-brain tissues were immediately collected. The brains were dissected, and small pieces of brain tissue were homogenized with 10 volumes of ice-cold PBS. The homogenates were directly centrifuged to obtain the supernatant to determine MDA levels (6000× *g* for 10 min at 4 °C) and AChE activity (14,000× *g* for 30 min at 4 °C). The MDA products were examined by monitoring thiobarbituric acid reactive substance formation. Each homogenate (160 μL) was mixed with 1% phosphoric acid (960 μL) and 0.67% thiobarbituric acid (320 μL). The mixture was incubated at 95 °C in a water bath for 1 h. After cooling, the absorbance of the colored complex was read at 405 nm. The protein concentration was determined using Bradford reagent (Bio-rad, Hercules, CA, USA).

### 3.5. Cholinergic System Activity

The small pieces of brain tissue were homogenated using a bullet blender (Next Advance Inc., Averill Park, NY, USA). The homogenates were centrifuged at 14,000× *g* for 30 min at 4 °C, and then the supernatant was used for the measurement of cholinergic system activity.

#### 3.5.1. AChE Activity

AChE activity was measured according to Ellman’s colorimetric method [53] using acetylthiocholine iodide as a substrate. The supernatant was mixed with a sodium phosphate buffer (50 mM, pH 8.0), and the mixtures were incubated at 37 °C for 15 min. After incubation, the mixtures were added to Ellman’s reaction mixture [1 mM 5,5′-dithio-bis(2-nitrobenzoic acid) and 0.5 mM acetylthiocholine iodide in a 50 mM sodium phosphate buffer (pH 8.0)] and incubated at 37 °C for 10 min. The absorbance was read at 405 nm using a microplate reader (Epoch2, BioTek, Winooski, VT, USA), and the results were expressed as percent relative to the activity of the control group (100%).

#### 3.5.2. ACh Content

An ACh level experiment was conducted according to the method of Vincent [54]. The supernatant was mixed with alkaline hydroxylamine reagent [2 M hydroxylamine in HCl and 3.5 N sodium hydroxide], and reacted at room temperature for 1 min. The mixture was added to 0.5 N HCl and 0.37 M FeCl_3,_ and then absorbance was immediately read at 540 nm using a microplate reader (Epoch2, BioTek, Winooski, VT, USA).

### 3.6. Mitochondrial Activity

#### 3.6.1. Isolation of Mitochondria from Brain Tissue

To isolate the brain mitochondria, whole brain tissue was added to five times the volume of the isolation buffer [215 mM mannitol, 75 mM sucrose, 0.1 % BSA, 20 mM HEPES (Na^+^), pH 7.2] with 1 mM EGTA, and homogenated using a bullet blender (Next Advance Inc., Averill Park, NY, USA). The homogenates were centrifuged at a low-speed spin (1300× *g* for 5 min), and the supernatant was transferred to new tubes and centrifuged again at 13,000× *g* for 10 min. The pellets were suspended in the isolation buffer with 1 mM EGTA and 0.1% digitonin in DMSO and incubated on ice for 5 min and were added to the isolation buffer with 1 mM EGTA and centrifuged at 13,000× *g* for 15 min. Next, the pellets were resuspended in isolation buffer and centrifuged at 10,000× *g* for 10 min, and the final pellet (isolated mitochondria) was suspended in isolation buffer (final protein concentration: 10 mg/mL).

#### 3.6.2. Mitochondrial ROS Content

Mitochondrial ROS content was evaluated using DCF-DA dye. Mitochondria isolated from brain tissue (final concentration: 0.8 mg/mL) were mixed with 25 μM DCF-DA in a KCl-based respiration buffer [125 mM potassium chloride, 1 mM magnesium chloride, 2 mM monopotassium phosphate, 5 mM pyruvate, 2.5 mM malate, 0.5 mM malate, and 20 Mm HEPES]. The mixture was incubated at room temperature for 20 min with aluminum foil. After incubation, fluorescence intensity was detected at an excitation wave (535 nm) and emission wave (485 nm).

#### 3.6.3. Measurement of MMP

MMP was measured using JC-1 dye, which could pass through the mitochondria membrane as a lipophilic material. Mitochondria isolated from brain tissue (final concentration: 1.2 mg/mL) were added to the assay buffer [5 mM pyruvate and 5 mM malate in isolation buffer with 1 mM EGTA] and the 1 μM JC-1 dye. The mixture was incubated at room temperature for 20 min with aluminum foil, and then fluorescence intensity was measured at an excitation wave (535 nm) and emission wave (590 nm).

#### 3.6.4. ATP Level

ATP was extracted with 1% TCA solution in ice for 10 min, and then 25 mM sodium acetate buffer (pH 7.8) was added until reaching pH 7.4. The ATP level was measured using a commercial kit (Promega, Madison, Wisconsin, USA) with a luminescence meter (Promega, Madison, WI, USA).

### 3.7. Western Blot Assay for Protein Expression 

Mouse brain mitochondria-mediated protein expression was measured using primary antibodies (BAX, cytochrome C). In brief, whole-brain tissue and isolated mitochondria were lysed with ProtinEx™ Animal cell/tissue (Gene All Biotechnology, Seoul, Korea) containing 1% protease inhibitor cocktails (Thermo Fisher Scientific, Rockford, IL, USA). Lysed tissue was centrifuged at 13,000× *g* for 10 min at 4 °C, and the protein content of the supernatant was measured using a Bradford reagent (Bio-rad, CA, USA). After that, 5× Laemelli buffer was added and boiled at 95 °C for 5 min. The proteins were separated in sodium dodecyl sulfate polyacrylamide gel electrophoresis and transferred to a polyvinylidene difluoride membrane (Millipore, Billerica, MA, USA). The membranes were blocked with 5% skim milk solution and incubated with primary antibody (1:1000) at 4 °C overnight. After incubation, the membrane was incubated with secondary antibodies (1:5000) for 2 h at room temperature. Finally, the bands were detected with ProNA™ ECL Ottimo (TransLab, Daejeon, Korea) using an iBright CL 1000 Imaging System (Thermo Fischer Scientific, Rockford, IL, USA).

### 3.8. Molecular Weight and Composition Analysis of Fucoidan from E. cava

#### 3.8.1. Determination of Sulfate and Monosaccharide Composition 

The sulfate content of fucoidan was determined using a BaCl2-gelatin solution. The fucoidan extract (3 mg) was added to 1 M HCl (5 mL) and heated at 105 °C for 5 h. Next, 3% TCA and BaCl_2_-gelatin solutions were mixed and incubated for 15 min. The released barium sulfate suspension was measured using a Beckman Coulter DU 650 spectrophotometer (Fullerton, CA, USA). The sulfate content was calculated with a standard curve of potassium sulfate [55].

The monosaccharide composition of fucoidan was determined using a high-pH anion-exchange chromatography with pulsed amperometric detection (HPAEC-PAD) system (Dionex, Sunnyvale, CA, USA) using a CarboPac^TM^ PA1 column (25 cm × 4 cm) with 18 mM NaOH for 15 min [56].

#### 3.8.2. Determination of Average Molecular Weight

The average molecular weight of fucoidan from *Ecklonia cava* was determined by gel permeation chromatography (GPC) using a Shodex SB 804 HQ OHPak column (30 cm × 8 cm) equilibrated with phosphate buffer (10 mM, pH 7) for 60 min. A pullulan polysaccharide calibration kit (Agilent Technologies, Santa Clara, CA, USA) was purchased and used as a molecular weight standard.

### 3.9. Statistical Analysis

All data were expressed as mean ± SD. The statistical significance of differences among groups was calculated by one-way analysis of variance (ANOVA). Significant differences were determined using Duncan’s new multiple-range test (*p* < 0.05) with SAS ver. 9.1 (SAS Institute Inc., Cary, NC, USA). 

## Figures and Tables

**Figure 1 marinedrugs-17-00591-f001:**
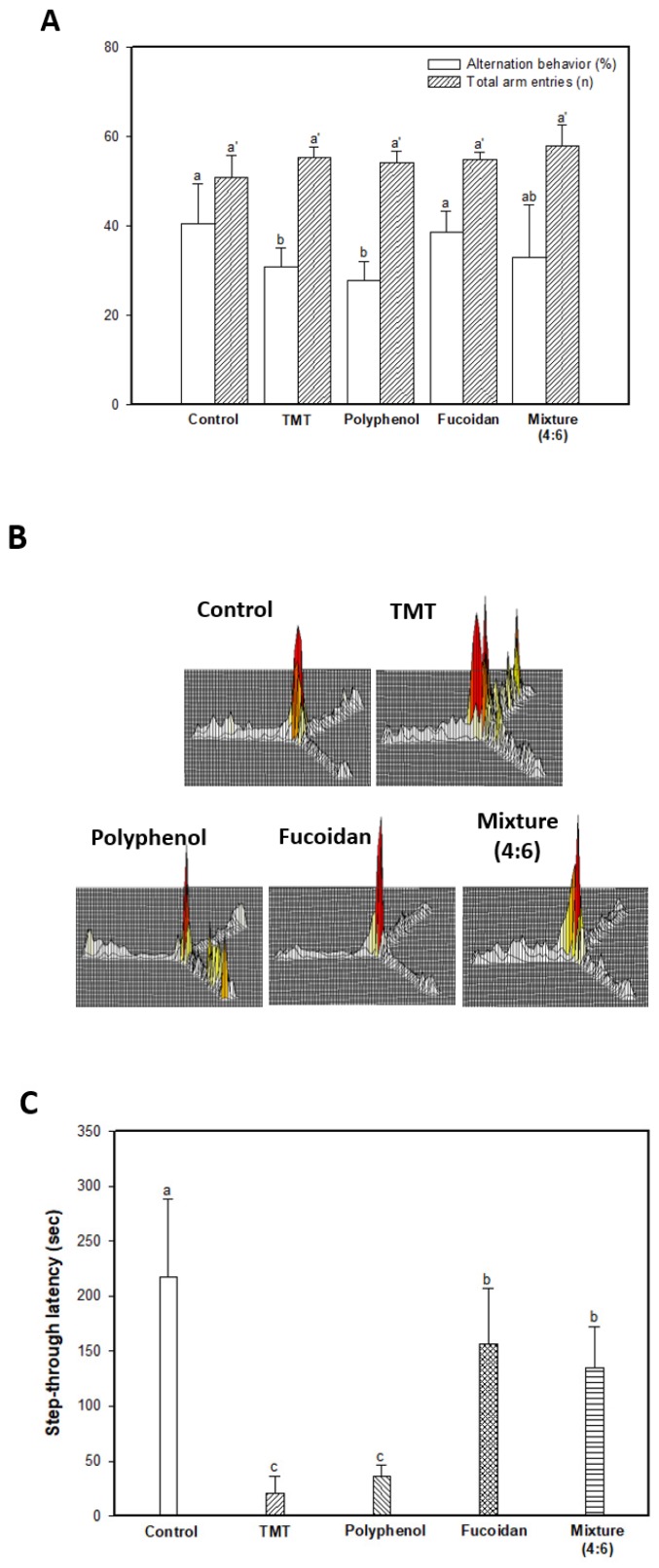
Ameliorating effect of polyphenol/fucoidan extract from *Ecklonia cava* and the mixture (4:6) in behavioral activity on TMT-induced learning and memory impairment mice. The spontaneous alteration behavior and number of arm entries (**A**) and path tracing of each group (**B**) in the Y-maze test and step-through latency time in the Passive avoidance test (**C**) were measured. Escape latency in the hidden-platform training trial (**D**), time in W zone for probe trial (**E**), and path tracing in the probe trial (**F**) in the Morris water maze test were also examined. The results were shown as means ± SD (*n* = 8), and were statistically considered at *p* < 0.05. Different small letters indicate a statistical difference.

**Figure 2 marinedrugs-17-00591-f002:**
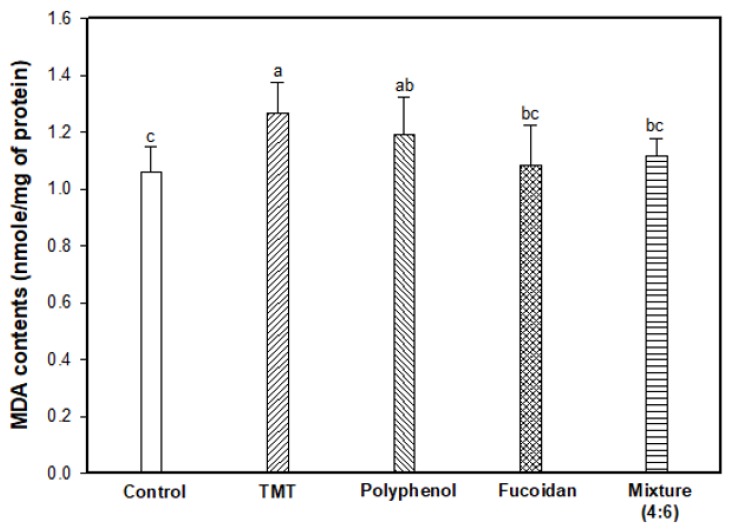
Inhibitory effect of polyphenol/fucoidan extract from *Ecklonia cava* and the mixture (4:6) against lipid peroxidation in brain tissue of TMT-induced cognitive dysfunction mice. The results were shown as means ± SD (*n* = 8) and were statistically considered at *p* < 0.05. Different small letters indicate a statistical difference.

**Figure 3 marinedrugs-17-00591-f003:**
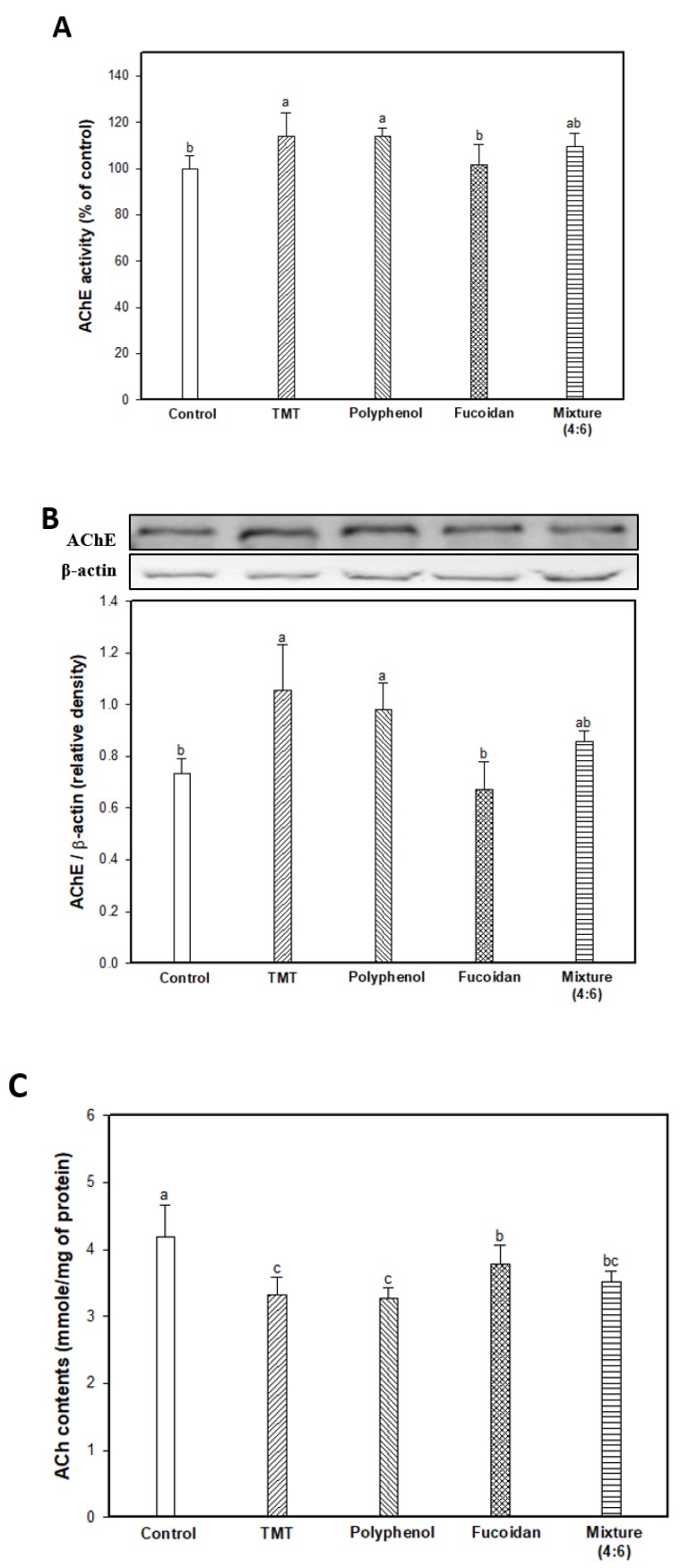
Effect of polyphenol/fucoidan extract from *Ecklonia cava* and the mixture (4:6) on TMT-induced cholinergic system dysfunction. AChE activity (**A**), band images of western blot analysis and AChE expression level (**B**), and ACh contents (**C**) in mouse brain tissue. The results were shown as means ± SD (*n* = 8) and were statistically considered at *p* < 0.05. Different small letters indicate a statistical difference.

**Figure 4 marinedrugs-17-00591-f004:**
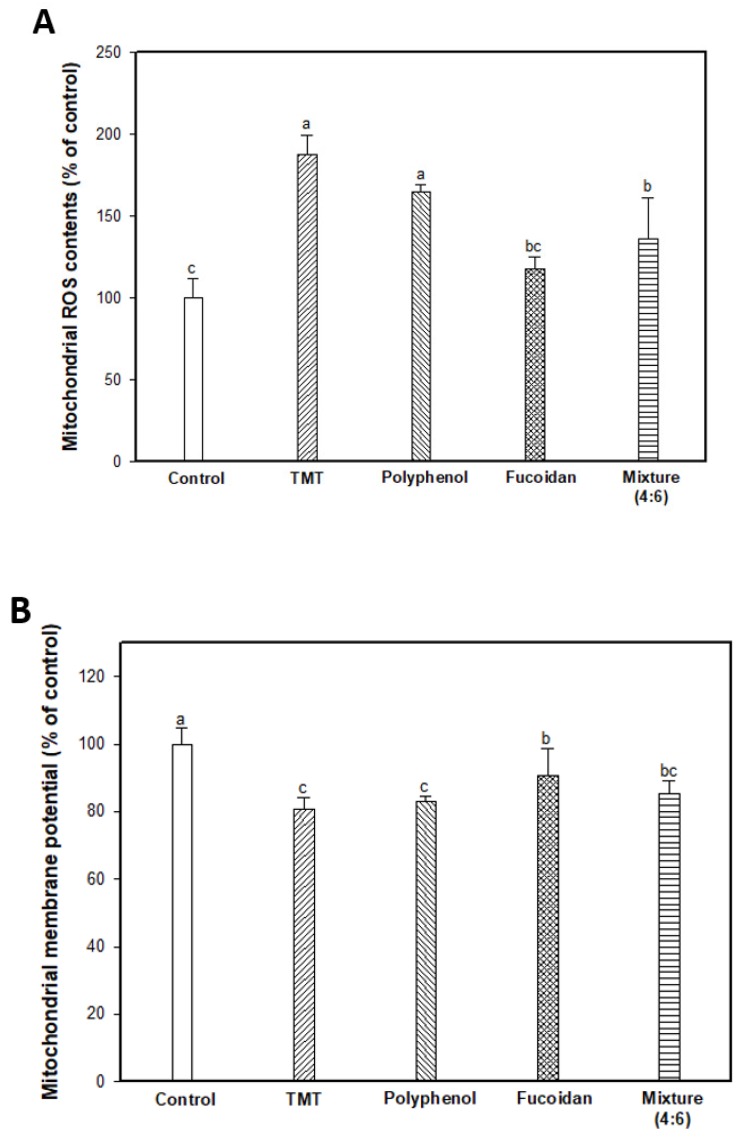
Mitochondrial activity assessments of polyphenol/fucoidan extract from *Ecklonia cava* and the mixture (4:6) on TMT-induced mitochondrial damage. Mitochondrial ROS contents (**A**), mitochondrial membrane potential (MMP) (**B**), ATP contents (**C**) in isolated mitochondria from brain tissue, band images of western blot analysis (**D**), the expression level of BAX (**E**), and cytochrome C (in mitochondria) (**F**) in mouse brain tissue. The results were shown as means ± SD (*n* = 5) and were statistically considered at *p* < 0.05. Different small letters indicate a statistical difference.

**Figure 5 marinedrugs-17-00591-f005:**
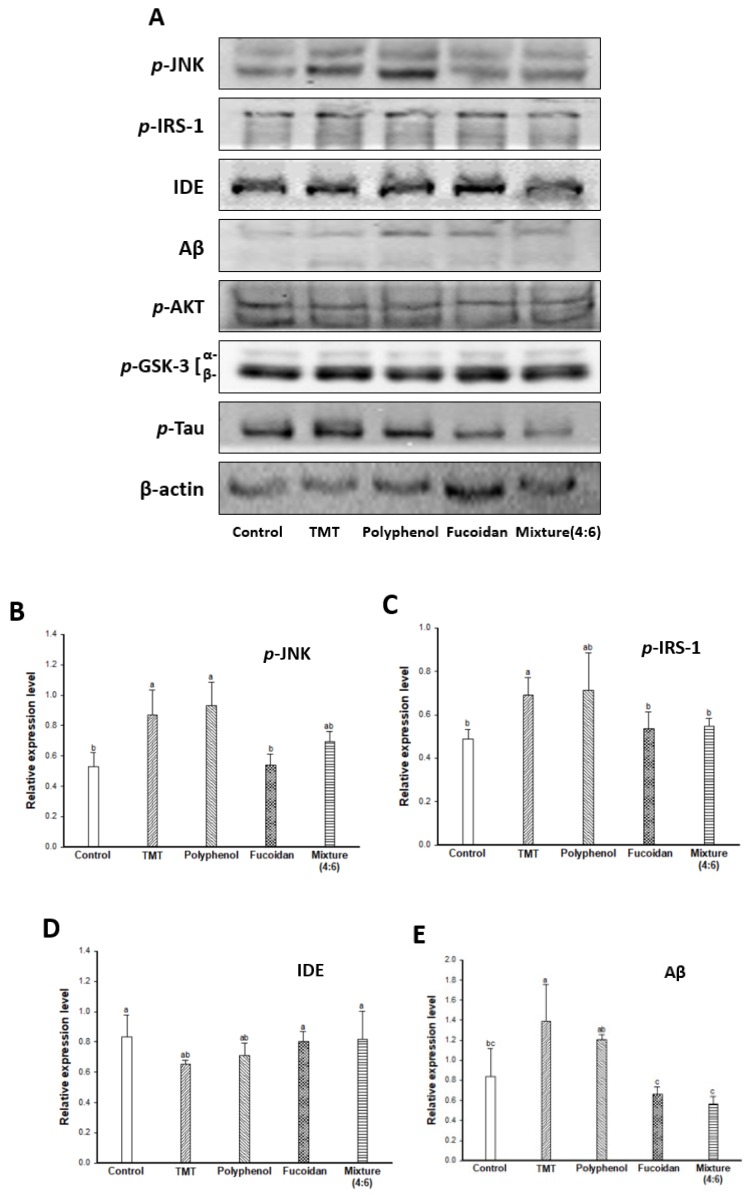
Effect of polyphenol/fucoidan extract from *Ecklonia cava* and the mixture (4:6) on TMT-induced cognitive dysfunction mouse model via down-regulation of amyloid β production/Tau hyperphosphorylation. Band image of western blot analysis (**A**), the expression level of *p*-JNK (**B**), *p*-IRS-1 (**C**), IDE (**D**), Aβ (**E**), *p*-AKT (**F**), *p*-GSK-3 (**G**) and *p*-Tau (**H**) in mouse brain tissue. The results were shown as means ± SD (*n* = 5) and were statistically considered at *p* < 0.05. Different small letters indicate a statistical difference.

**Table 1 marinedrugs-17-00591-t001:** Molecular weight and composition (w/w% of dried weight) of fucoidan extract from *Ecklonia cava.*

Mw (kDa)	Sulfate (%)		Monosaccharide Composition (Relative Area, %)						
			Arabinose	Fucose	Galactose	Glucose	Rhamnose	Xylose	Other
**110.78**	3.03%		2.03	15.19	9.62	11.42	1.44	23.77	36.53

**Table 2 marinedrugs-17-00591-t002:** Dieckol content (% (w/w)) of polyphenol extract from *Ecklonia cava* using the HPLC system.

Sample	Dieckol (% (w/w))
Polyphenol extract	4.8 ± 0.03

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
