# Peer review of "Fucoidan-Rich Substances from Ecklonia cava Improve Trimethyltin-Induced Cognitive Dysfunction via Down-Regulation of Amyloid β Production/Tau Hyperphosphorylation"

_marinedrugs, 2019, doi:10.3390/md17100591_

Round 1

Reviewer 1 Report

The manuscript by Seon Kyeong Park et al. is dedicated to the study of polyphenol: fucoidan = 4:6 extract on some on cognitive function. The paper should be revised before acceptance according to the next comments.

The methodological part have very limited information.

1.The object of study is not enough characterised. Authors must provide information about composition of polyphenol extract.

2. The yields of fucoidan and polyphenols extract should be indicated

3. Please provide references for the methods used in section 3.8.

4. Please provide the rationalities of doses/concentrations of polyphenol/fucoidan extract studied. Just single dose was studied. This is one of the main limitation of the study design.

- Fucoidan have high molecular weight and crossing of blood brain barrier by such molecules is questionable. In one of recent studies (https://doi.org/10.3390/md16040132) fucoidan was not observed in brains after per-oral administration. This aspect should be discussed in relation to effects of fucoidan studied on brain tissue homogenathes.

- The conclusion is not supported by data. No dose related effects has been studied. The study have very strong limitation and this should be indicated in the discussion and conclusion.

Reviewer 2 Report

It is an interesting paper. The Authors also looked at the cholinergic system examining AChE. I suggest to look also at the alpha7 nicotinic receptors since alpha7 is associated with Abeta and is involved in tauphosphorilation (see Crit Rev Toxicol 2012, 42:68; CurrDrug Targets 2017, 18:1179; Br J Pharmacol 2019, 176:3475). I recommend at least a western blotting.

Author Response

It is an interesting paper. The Authors also looked at the cholinergic system examining AChE. I suggest to look also at the alpha7 nicotinic receptors since alpha7 is associated with A-beta and is involved in tau phosphorylation (see Crit Rev Toxicol 2012, 42:68; CurrDrug Targets 2017, 18:1179; Br J Pharmacol 2019, 176:3475). I recommend at least a western blotting.

 → Thank you for your valuable suggestion. As your valuable recommendation, alpha 7 nicotinic acetylcholine receptor (α7nAchR) plays important role in inflammatory reaction as well as cholinergic system. If we follow your suggestions, we are confident that our research will be even more valuable. However, unfortunately, we do not have enough animal tissues to conduct the experiment. Several similar data can be found in recent researches. Therefore, if you understand, we will definitely reflect your suggestion in the next experiment. Thank you!

[Referenses]

Ha, J.S.; Jin, D.E.; Park, S.K.; Park, C.H.; Seung, T.W.; Bae, D.W.; Kim, D.O.; Heo, H.J. Antiamnesic effect of Actinidia arguta extract intake in a mouse model of TMT-induced learning and memory dysfunction. Evid.-based Complement Altern. Med. 2015, 2015, DOI: 10.1155/2015/876484.

Park, S.B.; Kang, J.Y.; Park, S.K.; Yoo, S.K.; Lee, U.; Kim, D.O.; Heo, H.J. Effect of Aruncus dioicus var. kamtschaticus extract on neurodegeneration improvement: ameliorating role in cognitive disorder caused by high-fat diet induced obesity. Nutrients, 2019, 11, DOI: 10.3390/nu11061319

Thanks for valuable suggestions!

Round 2

Reviewer 2 Report

For me the manuscript is suitable for publication in this form.